# The Relationship between Intolerance of Uncertainty and Problematic Social Media Use during the COVID-19 Pandemic: A Serial Mediation Model

**DOI:** 10.3390/ijerph192214924

**Published:** 2022-11-13

**Authors:** Chaoran Sun, Yumei Li, Sylvia Y. C. L. Kwok, Wenlong Mu

**Affiliations:** 1Department of Behavioural and Social Sciences, City University of Hong Kong, Tat Chee Avenue, Kowloon, Hong Kong SAR 518057, China; 2School of Journalism and Communication, Wuhan University, Bayi Road, Wuchang District, Wuhan 430072, China

**Keywords:** maladaptive coping strategies, intolerance of uncertainty, fear of missing out, problematic social media use

## Abstract

The COVID-19 pandemic has brought significant interruptions to life certainty, and there has been a lack of research on the influence of uncertainty. The present research aimed to explore how intolerance of uncertainty, maladaptive coping strategies, and fear of missing out affect social media use in a Chinese community sample (N = 311) during the pandemic. Serial mediation analysis was applied, integrating the mediating role of maladaptive coping strategy and fear of missing out. Intolerance of uncertainty, maladaptive coping strategies, and fear of missing out was positively related to PSMU. Based on the mediation analysis, when age and gender were controlled, the direct effect of intolerance of uncertainty on PSMU was significant. The total indirect effect was also significant. The effect of intolerance of uncertainty on PSMU was mediated by maladaptive coping strategies and fear of missing out. Taken together, maladaptive coping strategies and fear of missing out played a serial mediating role between intolerance of uncertainty and PSMU. The findings imply that strategies to improve the tolerance of uncertainty, reduce fear of missing out, and relevant coping strategies could be potentially helpful in mitigating problematic social media use, especially during the COVID-19 pandemic.

## 1. Introduction

Social media has become indispensable in recent years, especially since the onset of the novel coronavirus disease 2019 (COVID-19). A huge amount of in-person communication was moved online, especially during strict lockdown periods [1]. Social media is also referred to as social networking sites where people chat and share information with friends, acquaintances, and new people [2]. Social media has a crucial role in disseminating information, tackling misinformation, and advocating for health decisions, in some ways providing certainty and relief in a fluctuating period such as the COVID-19 pandemic [1,3,4,5,6]. The back-and-forth lockdown policy has led to individuals being confined to sedentary behaviors, where it is easy to browse social media for an excessive amount of time [7]. Under this context, the behavior of problematic social media use (PSMU) has drawn researchers’ attention. Signs suggesting PSMU include experiencing positive mood online and mood decline when offline; substantial period spent on using social media; increasing amount of time to achieve gratification; failure to restrict the time online and feeling distressed when failing to do so; replacing in-person social activities with social media; and a tendency to return to previous behavioral patterns after a period of abstinence or regulation [8]. PSMU was found related to increased attention deficit hyperactivity symptoms over time [9] and the deterioration of mood disorders (e.g., depression) [10]; it was also related to undermined job performance [11].

With many scholars studying the psychopathological mechanism behind PSMU, recent studies began to investigate the underlying transdiagnostic factors [12]. “Transdiagnostic” refers to the core vulnerability that overlaps multiple psychological disorders [13]. Intolerance of uncertainty (IU), a transdiagnostic factor, was positively correlated to nomophobia, the fear when the mobile phone is out of sight [14]. IU refers to the unacceptance of the fluctuating likelihood of negative events, making the individual distressed by the pressure induced by the uncertainty [15]. The compensatory internet use theory [16] states that the reliance on specific platforms, e.g., the internet, is to compensate for the needs unmet in real life. The empirical literature proves that the purpose of addictive behavior is to relieve negative feelings or as a distraction from distress [17]. In accordance with the compensatory internet use theory, feeling distressed by uncertainties offline may make people rely on social media to escape from the stress and try to find certainty via social media use. However, little studies investigated the relationship between IU and PSMU. In a meta-analysis covering articles from 1999 to 2014 [18], the general public’s intolerance of uncertainty has escalated, and so has the use of mobile phone. In a two-wave study before the pandemic, Rozgonjuk, Elhai, Täht, Vassil, Levine and Asmundson [12] discovered that IU was positively correlated to problematic smartphone use. In the COVID-19 pandemic, IU was found to be associated with the fear of coronavirus [19], so it is likely that individual’s IU could be salient amid the pandemic. According to the survey data of three hundred participants in Turkey, IU was positively correlated to nomophobia, the fear when the mobile phone is out of sight [20]. Meanwhile, the mechanism of PSMU was found to largely overlap with the smartphone use issue [21]. Therefore, it is reasonable to posit a positive association between IU and PSMU (Hypothesis 1).

Although there is evidence indicating that IU is correlated with PSMU, the mechanism of how IU links to PSMU needs further exploration. Based on the Interaction of Person-Affect-Cognition-Execution (I-PACE) model [22,23], problematic technology use is attributed to the interaction among personal characteristic variables, affective and cognitive reaction to specific triggers, and executive functions. In the I-PACE model, if the sources of behavior belong to part of an individual’s core dispositions, it might lay the foundation for developing PSMU. Herein, IU has a role as a disposition that could have a deep impact on PSMU. Other than personal core disposition, the cognitive response is one of the mediating variables explaining problematic use. The term ‘coping strategy’ refers to the cognitive or behavioral level effort to address the issues brought by a stressful event [24]. Evidence suggests that those with high IU prefer maladaptive coping mechanisms such as behavioral disengagement, avoidance, and substance reliance [25]. Maladaptive coping strategies, especially behavioral disengagement [17] and avoidance [26], were positively and significantly related to the time spent online. Meanwhile, previous literature shows that maladaptive coping strategies partially mediated the relation between video game addiction and declined mental health [27]. In the COVID-19 pandemic, maladaptive coping strategies was found as a mediator of the connection between IU and psychological distress, such as generalized anxiety and depressive symptoms [28]. In a repeated measurement study amid the pandemic, maladaptive coping strategies served as a mediator between distress intolerance and problematic internet use [29]. Thus, it is proposed that maladaptive coping strategies mediate the relationship between IU and PSMU (Hypothesis 2a).

Except for maladaptive coping strategies, the previous literature also showed that fear of missing out is a psychological issue closely related to PSMU. Fear of missing out is defined as the apprehension that other people might be enjoying their lives without one’s presence, which results in the tendency of staying consistently updated with what others are doing [30]. Through the self-determination theory, when one’s basic psychological needs cannot be satisfied, they utilize a self-regulatory state of limbo such as fear of missing out as a means of remediation [30,31]. It was found that the fear of missing out has played a mediating role in problematic social networking site use in the COVID-19 lockdown in Italy [32]. Similarly, individuals who are looking for certainty (i.e., people with high IU) may also be prone to stay connected with what others are doing to obtain certainty. Hence, it could be proposed that there is a mediation effect of fear of missing out on the association between IU and PSMU (Hypothesis 2b).

Considering the nature of maladaptive coping strategies that enable the individual to avoid confronting the feared situation to increase apprehension [33], there is likely a serial mediation between maladaptive coping strategies and fear of missing out. According to the I-PACE model [23], problematic technology use is due to the interaction between pre-dispositional factors, affective responses, cognitive responses, and executive behavioral responses. Examples of cognitive responses include coping strategies [34] and maladaptive cognitive biases such as fear of missing out [35,36,37]. Maladaptive coping strategies, such as avoidant ones, were positively correlated with worry [38]. While in the study, worry was defined as a cognitive behavior and problem-solving tendency to avoid threats, fear of missing out has a similar mechanism to avoid the unpleasant feelings via undertaking the strategy of staying connected to others’ posts on the social media. Additionally, fear of missing out is related to a deficiency in adaptive coping strategies [39]. Thus, the extant literature indicates that both fear of missing out and maladaptive coping strategies are closely related to PSMU, but comparable studies have not yet assessed that whether the two would relate to PSMU in a sequential manner. Therefore, the present study would test a serial mediation model of maladaptive coping strategies and fear of missing out in relation to IU and PSMU in the context of COVID-19 (Hypothesis 3).

In summary, given the research gap in the relationship between IU, maladaptive coping strategy, fear of missing out, and PSMU, the present research aimed at eliminating the research gap via examining the association between the four variables. Specifically, the present research focused on exploring the mediating roles of maladaptive coping strategies and fear of missing out on the relationship between IU and PSMU. The findings would benefit the understanding of the association between IU and PSMU.

## 2. Materials and Methods

### 2.1. Participants and Procedures

The current study used a cross-sectional design. A convenient sampling method was adopted to recruit participants from communities and colleges in China. Ethics approval has been obtained from the College Human Subjects Ethics Sub-Committee of College of Liberal Arts and Social Sciences of City University of Hong Kong (2020-21-CIR8-3). Data collection was conducted in April 2021. Given that it was during the coronavirus pandemic, an online survey was adopted to collect the data. The survey link was distributed via WeChat, QQ, and other commonly used social platforms in Mainland China. Participant inclusion criteria were: (1) older than 18 years old; (2) native Chinese speaker; (3) people with experience in using social media. The data exclusion criteria were: (1) did not complete the whole survey; (2) provided unconventional answers to the polygraph items, which asked the participants to re-indicate their gender during the survey. The online informed consent of participants was obtained. All participants were aware of the voluntary nature of the study. The data were anonymous and have been kept strictly confidential. Only the research team members of the present study have the access to the data. A total of 378 participants were recruited from 30 provincial districts. Overall, 50 participants were excluded because they did not submit their questionnaire responses. In total, 27 were excluded because they failed the polygraph items. Hence, 311 participants (response rate = 82.28%) provided complete and valid (without unconventional responses on the polygraph items) answers. The sample mean age was 28.67 years (SD = 7.72), ranging from 18 to 54. Among the participants, 188 (60.45%) were female, and 123 (39.55%) were male. In total, 135 (43.41%) were single, 101 (32.48%) were married, 70 (22.51%) were in a relationship, and 5 (1.61%) were divorced. A total of 53 (17.04%) participants did not attend university, whereas 258 (82.96%) achieved their bachelor’s or higher.

### 2.2. Measures

#### 2.2.1. Intolerance of Uncertainty

The 12-item Intolerance of Uncertainty Scale [40] was adopted to measure responses and beliefs to uncertainty, ambiguity, and impending situations. The scale has two factors (i.e., prospective IU and inhibitory IU), and each factor includes six items. The sample items are “Unforeseen events upset me greatly” (prospective IU) and “Uncertainty keeps me from living a full life” (inhibitory IU). Participants were required to respond on the 5-point Likert scale ranging from 1 (Not at all true of me) to 5 (Extremely true of me). A higher total score indicates that the participants had a greater extent of intolerance of uncertainty. The Chinese version scale showed good reliability and validity among the Chinese-speaking adults [41]. The Cronbach’s alpha of the scale in the present study was 0.953.

#### 2.2.2. Maladaptive Coping Strategies

The behavioral disengagement, self-blame, substance use, denial, and self-distraction subscales (two items per subscale) in the Brief COPE Inventory [42] were used to evaluate the maladaptive coping strategies. A sample item is “I’ve been blaming myself for things that happened.” Participants needed to rate each item on a 4-point Likert scale ranging from 0 (I haven’t been doing this at all) to 3 (I’ve been doing this a lot). A higher total score demonstrates that the participants are more likely to adopt maladaptive coping strategies. A previous study suggests a good psychometric characteristic of Brief COPE among the Chinese population [43]. The Cronbach’s alpha of the scale in the present study was 0.831.

#### 2.2.3. Fear of Missing Out

Fear of missing out was evaluated by the 10-item Fear of Missing Out Scale [30]. A sample item is “I get anxious when I don’t know what my friends are up to”. Participants were asked to rate on a 5-point Likert scale ranging from 1 (Not at all true of me) to 5 (Extremely true of me). A high mean score demonstrates that the participants had a higher level of apprehension and fear of missing out. In a previous study, the Chinese version of the Fear of Missing Out Scale was validated among adolescents and emerging adults [44]. The Cronbach’s alpha of the scale in the present study was 0.905.

#### 2.2.4. Problematic Social Media Use

Problematic social media use was measured by the 6-item Bergen Social media Addiction Scale [45]. A sample item is “Use a lot of time thinking about or planning using social media”. Participants were required to respond on the 5-point Likert scale ranging from 1 (Very rarely) to 5 (Very often). A higher total score suggests that participants exhibit more problematic social media use behaviors. The Chinese version scale had been validated among adults and adolescents [46,47]. The Cronbach’s alpha of the scale in the present study was 0.914.

### 2.3. Statistical Analysis

All the analyses were conducted in SPSS 26.0. First, Harman’s single factor test with principal component analysis was applied to test for common method bias. Descriptive statistics, skewness, kurtosis, and Pearson bivariate correlations were calculated among the key variables. Absolute skewness ≤ 2.0 and/or absolute kurtosis ≤ 7.0 were adopted as the criteria of data normality [48]. Second, two simple mediation analyses were performed using model 4 in the PROCESS macro. Third, the serial mediation model was tested using model 6 in the PROCESS. Maladaptive coping strategies were set as the first mediator, and fear of missing out was set as the second mediator. Gender and age were controlled in all the mediation models, because gender and age were proved to be confounding variables in relation to PSMU [49,50]. The 95% confidence interval (CI) estimated using 5000 random bootstrapping does not contain zero was adopted as the criteria for the significance of the mediation effect [51].

## 3. Results

### 3.1. Preliminary Analyses

Harman single-factor test results showed that six unrotated factors accounting for 67.65% of the total variance. The first single factor accounted for 37.89% of the variance, less than the critical 40% threshold [52]. Hence, the common method bias was not a big concern in the current study.

Means, standard deviations, kurtosis, skewness, and correlation coefficients of the research variables were presented in Table 1. Skewness and kurtosis results showed no departure from normality in the current data. IU was significantly and positively correlated with fear of missing out, maladaptive coping strategies, and PSMU. Maladaptive coping strategies were positively and significantly correlated with fear of missing out and PSMU. The correlation between fear of missing out and PSMU was also significant.

### 3.2. The Mediating Roles of Maladaptive Coping Strategies and Fear of Missing Out

The current study applied model 4 in the PROCESS macro to test the indirect effect of maladaptive coping strategies in the connection between IU and PSMU. IU was significantly related to maladaptive coping strategies and PSMU. Maladaptive coping strategies showed a positive and significant association with PSMU. The whole model was significant and accounted for 60.50% of the variance in PSMU. As shown at the top of Table 2, the direct effect of IU on PSMU was significant in the model. Maladaptive coping strategies were a mediator in the association between IU and PSMU in the model.

Model 4 in the PROCESS was also adopted to analyze the indirect effect of fear of missing out in the association between IU and PSMU. IU was positively associated with fear of missing out. Fear of missing out was significantly and positively correlated with PSMU. The whole model was significant and accounted for 49.05% of the variance in PSMU. The lower part of Table 2 showed that the direct effect of IU on PSMU was significant in the model. Fear of missing out mediated the association between IU and PSMU in the model.

### 3.3. Exploring the Serial Mediation Model

The PROCESS macro, model 6, was applied to test the serial mediating effects of maladaptive coping strategies and fear of missing out on the association between IU and PSMU. The analysis results are shown in Figure 1 and Table 3. IU was significantly related to maladaptive coping strategies, fear of missing out, and PSMU. Maladaptive coping strategies were significantly and positively linked to fear of missing out as well as PSMU. The positive effect of fear of missing out on PSMU was significant. The whole model was significant and accounted for 67.34% of the variance in PSMU. There was a significant direct effect of IU on PSMU. The total indirect effect was significant. The effect of IU on PSMU was mediated by maladaptive coping strategies and fear of missing out. The two mediators sequentially mediated the path from IU to PSMU.

## 4. Discussion

The COVID-19 pandemic has brought significant interruptions to certainty, and there has been a research gap on the association between IU and PSMU. The present study aimed at adding to the understanding of the mechanisms between IU and PSMU during the COVID-19 context, via testing a serial mediation model. In accordance with Hypothesis 1, the result showed plausible evidence that higher IU was directly correlated with more maladaptive coping strategies, fear of missing out, and PSMU. The simple mediation analysis indicated that the indirect effect between IU and PSMU could be explained by fear of missing out and maladaptive coping strategies, supporting Hypothesis 2a and Hypothesis 2b. Particularly, the current research supported the serial mediation model in which maladaptive coping strategies and fear of missing out mediated the relationship between IU and PSMU in a sequential manner, supporting Hypothesis 3.

According to the I-PACE model, personal core dispositions could prevent or induce internet-related problems. IU has been recognized as one of the vulnerabilities for developing addiction issues [53] and was also found to be one of the predictors of internet addiction after the outbreak of COVID-19 [54,55]. Furthermore, PSMU could be a compensation for psychological fulfillment when there is a lack of intended stimulus in real life [16]. Therefore, due to the excessive uncertainties brought by the pandemic, people with higher IU have turned to social media more often to seek relevant news and announcement from the government officials to secure certainties. The present research is consistent with this statement as a significant direct effect between IU and PSMU was found, supporting Hypothesis 1. The current results are in accordance with previous studies [12,56].

Results in the current study indicated that maladaptive coping strategies mediated the relation between IU and PSMU. Previously, researchers have found the association between IU and the development of maladaptive coping strategies [57,58]. Possible explanation to the present finding is that IU is one of the predictors of anxiety symptoms and heightened worry [59], whereas those who have severe anxiety usually prefer maladaptive coping strategies [60,61]. It is also likely that people who prefer maladaptive coping strategies have an escape coping motive, so they are more easily to rely on the online activities [62]. The current study suggested that fear of missing out was a mediator between IU and PSMU. Fear of missing out is characterized as the desire to maintain ongoing contact with others. Fear of missing out could be more serious during the pandemic due to the loss of face-to-face contact and opportunities to establish secure relationships. Under this context, it is possible that individuals who are intolerant to uncertainty have more health-related anxiety [63,64,65], and thus more easily worry about themselves and others. Fear of missing out could be increased along with worry; thus, people turn to social media in order to stay updated to the situation of others, given that in-person social activities were confined during the pandemic [37,66].

In addition, the present study tested a serial mediation model behind IU and PSMU. The results aligned with the underlying mechanism proposed by the I-PACE model, that factors associated with problematic technology use issues are inter-related [22,23]. The current results demonstrated that PSMU is a multi-dimensional issue, and it needs to be interpreted with the related transdiagnostic factors, cognitive and emotional factors behind. Our results were consistent with previous findings. The maladaptive coping strategies was found to be correlated with the reduction in subjective well-being during the COVID-19 pandemic [67,68,69]. Previous literature suggested that those who reported higher levels of well-being tend to have less attachment to their smartphone, and experienced less fear of missing out [70]. Because fear of missing out was manifested as hindered self-regulation, people who are unable to cope with the distress brought by the uncertainties may seek other accessible platforms such as social media to ease the negative feelings brought by the pandemic [66]. Therefore, as stated in the results of the current study, people who cannot endure uncertainty may prefer maladaptive coping strategies, which undermine well-being and life satisfaction, then produce heightened fear of missing out and lower self-control. Self-control was negatively correlated with problematic technology use issue [71]. All these factors make the individuals feel more dissatisfied in life, so they turn to social media platforms for temporary fulfillment.

To the authors’ knowledge, this is one of the first studies to assess relationships between IU, maladaptive coping mechanisms, fear of missing out, and PSMU during the COVID-19 pandemic. The findings provided several implications to reduce PSMU. Though behavioral intervention such as setting a limit on screen time has been popular [72], the issues induced by IU, fear of missing out, and maladaptive coping strategies may become barriers. For instance, IU as a transdiagnostic construct has been a significant hindrance in delivering exposure therapy, according to a meta-analysis [73]. It is known that problematic technology use serves as a means for the individual to gain gratification [74], so it is important to open easy access to alternative sources of gratification in the lockdown period, such as recordings on easy exercise at home. Even though IU is regarded as a stable characteristic, evidence shows that building cognitive and behavioral coping strategies could be helpful to uncertainty management [75]. Educational and clinical researchers may find that training the tolerance of uncertainty could help the individuals stick to adaptive coping strategies and reduce the fear of missing out. Drawing from the compensatory internet use theory, the compensatory role of social media could thus be replaced. Notwithstanding the paucity of mental health intervention targeting fear of missing out, empirical evidence revealed that time management and sleep hygiene intervention were effective in weakening the fear of missing out on PSMU [76]. When developing prevention strategies, researchers should be encouraged to consider the relationship between IU and PSMU, and incorporating techniques to target IU, maladaptive coping strategies and fear of missing out.

The results of the present research need to be interpreted considering some limitations. First, even though we proposed a theory-based model, we were not able to provide causality inference due to the cross-sectional nature of the current research. A bidirectional association might exist between IU and PSMU. The increased engagement with mobile phones helps individuals to seek reassurance and quickly gain certainty. As a result, the individual’s tolerance for uncertainty gradually decreases, i.e., IU increases [18]. Hence, future studies are warranted to incorporate experiments and longitudinal design to assess the causality. Secondly, more than half of our participants received a college education or above and/or are Generation Z (206 of our valid answers were from participants who were born in 1990–2010). Given that Gen Z may have their own values and criteria of dealing with issues [77], the generalizability toward individuals with lower education levels and older generation is limited. Future studies may consider broadening the sample recruitment to cover the underrepresented population, for instance, those with lower educational background and those in middle and late adulthood. Third, we utilized the self-report measurement, which is susceptible to recall bias or social desirability bias. Behavioral experiment could be one of the future directions to complement this limitation. Furthermore, although IU, fear of missing out, and maladaptive coping strategies are closely connected with psychopathologies (e.g., anxiety symptoms), the current study did not screen the psychopathology history of the participants. Hence, future research could incorporate such screening to exclude the influence of psychopathology on PSMU.

## 5. Conclusions

The relationship between the transdiagnostic construct and PSMU has been understudied, making it particularly important to explore the relationship between IU and PSMU, especially in the context of the great uncertainty that COVID-19 brings to society. The present study demonstrated the mediating role of maladaptive coping strategies and the fear of missing out by constructing a serial mediation model between IU and PSMU. The current study extends the application of the I-PACE model to the field of PSMU and enriches the research literature on the relationship between IU and PSMU. In addition, the results of this study provide insight and empirical evidence for strategies that consider reducing IU, maladaptive coping strategies, and fear of missing out on PSMU prevention. The research result needs to be interpreted considering the limitation of the cross-sectional nature, generalizability, and the potential influence of psychopathology.

## Figures and Tables

**Figure 1 ijerph-19-14924-f001:**
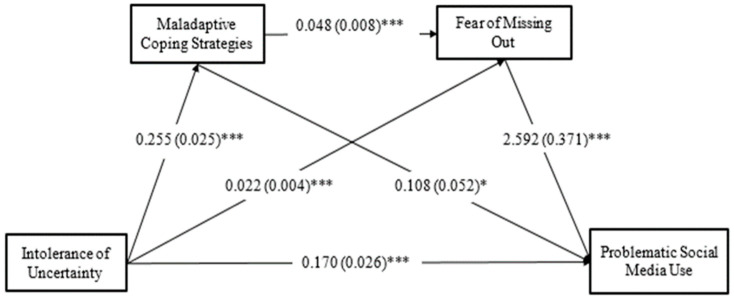
Serial mediation of the relation between intolerance of uncertainty and problematic social media use by the maladaptive coping strategies and fear of missing out. Age and gender were controlled during the data analysis but are not presented for reasons of simplicity. *** *p* < 0.001, * *p* < 0.05.

**Table 1 ijerph-19-14924-t001:** Means, standard deviations, skewness, kurtosis, and correlations of the study variables (*n* = 311).

	Variables	Intolerance of Uncertainty	Maladaptive Coping Strategies	Fear of Missing Out	Problematic Social Media Use
1	Intolerance of Uncertainty	1			
2	Maladaptive Coping Strategies	0.516 ***	1		
3	Fear of Missing Out	0.489 ***	0.507 ***	1	
4	Problematic Social Media Use	0.566 ***	0.462 ***	0.581 ***	1
	Mean	35.72	14.49	3.31	19.12
	SD	11.27	5.70	0.789	5.66
	Skewness	−0.088	−0.079	−0.600	−0.351
	Kurtosis	−1.041	−0.482	−0.047	−0.508

Note. M = mean, SD = standard deviations; *** *p* < 0.001.

**Table 2 ijerph-19-14924-t002:** Bias-corrected bootstrap test in mediating effect when maladaptive coping strategies/fear of missing out was entered as the mediation variable (*n* = 311).

Model Pathways	Effect	95% Boot CI
Direct Path		
IU → PSMU	0.226	[0.174, 0.279]
Indirect Path		
IU → Maladaptive Coping Strategies → PSMU	0.059	[0.030, 0.095]
Direct Path		
IU → PSMU	0.188	[0.140, 0.237]
Indirect Path		
IU → Fear of Missing Out → PSMU	0.097	[0.058, 0.143]

Note. IU = Intolerance of Uncertainty, PSMU = Problematic Social Media Use.

**Table 3 ijerph-19-14924-t003:** Bias-corrected bootstrap test in mediating effect when maladaptive coping strategies and fear of missing out were entered as the mediation variables (*n* = 311).

Model Pathways	Effect	95% Boot CI
Direct Path		
IU → PSMU	0.170	[0.118, 0.221]
Indirect Path		
Total: IU → PSMU	0.116	[0.070, 0.168]
IU → Maladaptive Coping Strategies → PSMU	0.028	[0.002, 0.059]
IU → Fear of Missing Out → PSMU	0.057	[0.025, 0.095]
IU → Maladaptive Coping Strategies → Fear of Missing Out → PSMU	0.032	[0.015, 0.054]

Note. IU = Intolerance of Uncertainty, PSMU = Problematic Social Media Use.

## Data Availability

The datasets generated during and/or analyzed during the current study are available from the corresponding author on reasonable request.

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
