# Peer review of "The Relationship between Intolerance of Uncertainty and Problematic Social Media Use during the COVID-19 Pandemic: A Serial Mediation Model"

_ijerph, 2022, doi:10.3390/ijerph192214924_

Round 1

Reviewer 1 Report

Dear authors,

The article is interesting and works in depth on the object of study: exploring how intolerance to uncertainty, maladaptive coping strategies and fear of missing out affect the use of social networks. The findings are timely for the scientific community and for society at large.

v  The title is appropriate and beautiful. It clearly states the claims of the article.

v  The abstract presents the objectives of the study in an orderly manner. There is a brief introduction and the data, results and conclusions are presented in an appropriate manner.

v  The keywords are appropriate.

v  The introduction provides a multitude of previous studies that provide insight into how the pandemic and problematic use of social networks are related to mood and dependence. In addition, all the articles cited are highly topical.

§  It is correctly detailed that the fear of missing out links the deficits in psychological needs to PSMU.

§  The hypotheses are sound and easy to address.

§  The aim of this research is well reflected: The present study aimed at testing a serial mediation pathway of maladaptive coping strategies and fear of missing out in relation to IU and PSMU in the COVID-19 context.

v  Materials and methods conform to what is desirable for research of this type.

§  The procedure and participants are well detailed. The socio-demographic data of the sample are described rigorously.

§  The instrument is well described and the essential information for each scale or subscale is detailed in each section. Cronbach's alpha values are provided.

§  The 'statistical analysis' section contains relevant information.

v  The results section provides sufficient tables with useful and necessary content.

§  The results are presented in an orderly manner, so that the findings are presented in a step-by-step manner. These findings are presented in a rigorous manner.

§  IU was significantly and positively correlated with fear of missing out, maladaptive coping strategies, and PSMU. Maladaptive coping strategies were positively and significantly correlated with fear of missing out and PSMU. The correlation between fear of missing out and PSMU was also significant.

§  As shown at the top of Table 2, the direct effect of IU on PSMU was significant in the model. Maladaptive coping strategies were a mediator in the association between IU and PSMU in the model.

v  The discussion relates to the previous studies provided in the "introduction" section and reflects the relationship with the results found.

§  Interestingly, the authors indicate that: this is one of the first studies to assess relationships between IU, maladaptive coping mechanisms, fear of missing out, and PSMU under the COVID-19 pandemic.

§  Limitations are well presented.

§  It is presented as different prospective research to resolve some limitations of the study and to answer new questions that have arisen.

v  The conclusions, although brief, capture very well the purpose of this study and the reflection of the results obtained on tolerance to uncertainty, fear of missing out and coping mechanisms.

v  The references are well done, up to date and in line with the standards set by MDPI.

However, I offer some suggestions for improvement:

1.      Introduction: when you write "et al", it is not usually italicised. I recommend that you remove italics from the entire document. Nor should "et al" be preceded by ",".

2.      Section 2.1. It is necessary to indicate that this study has the approval of the Human Research Ethics Committee, as specified in lines 349-351. It is pleasant to understand from the outset that this research received a favourable report to be carried out.

3.      Section 2.1. It is necessary to indicate whether the participants were informed of the voluntary nature of the study, the exclusivity of the data and the confidentiality of the data.

4.      What do the numbers in the first row of Table 1 mean? I think this should be better explained or even a note should be added at the end of the table.

5.      The title of the tables should be centred, according to the MDPI guidelines.

6.      Figure 1 could be centred with the text instead of being left-justified.

7.      The aim of the study seems confusing at section "4. Discussion". Lines 120 and 121 state a different objective than the one described in lines 244-245. I believe that what is described in lines 244-245 is one of the objectives of the study or even the purpose itself.

8.      References: it is necessary to indicate the link to the "doi". It is more correct to put at the end of the reference: https://doi.org/10.1016/S2589-7500(20)30315-0  instead of "doi: 10.1016/S2589-7500(20)30315-0". All references including doi need to be corrected.

I recommend that these recommendations for improvement be heeded. I am sure they will improve the quality of this research article.

Author Response

Dear reviewer,

I hope you are doing well. Please see the attachment for the response.

Thank you for your time and help,

Chaoran

Reviewer 2 Report

Please indicate the research gap and the method of its elimination by the authors already in the abstract. You wrote: „Intolerance of uncertainty, maladaptive coping strategies, and fear of missing out were positively related to PSMU.” It should be:  “was”, not “were”. Please replace it. You wrote: “helpful to mitigate problematic social media use, especially under the COVID-19 pandemic”. The word “to mitigate” does not seem to work here. Please consider the word “in mitigating”. You wrote: “especially under the COVID-19 pandemic”. It seems that the preposition used may be incorrect here. Please consider the word “during”. ….”could be potentially helpful in mitigating problematic social media use, especially during the COVID-19 pandemic.” I encourage the authors to reflect on the correctness of the English text. Occasionally, some discrepancy in the use of certain words may render the text incomplete. 

Please consider the use of research on the role of framing in the fight against Covid. It is also part of social media. See the article "The Strategy of Vaccination and Global Pandemic: How Framing May Thrive on Strategy During and After Covid-19"  and other articles. Please expand the scope of your literature studies to include Covid from an organizational countermeasure perspective. Social media is a valuable preventive tool. See the article "Why Some Countries Win and others Lose from the COVID-19 Pandemic? Navigating the Uncertainty" and other articles. Please also pay attention to the fact that representatives of Gen Z use social media (see the article "Understanding the Impact of Generation Z on Risk Management—A Preliminary Views on Values, Competencies, and Ethics of the Generation Z in Public Administration" and other articles). This fact is significant because Gen Z's dependence on social media is greater than in the case of older people. Please complete the literature studies with Gen Z and indicate in "Materials and Methods" how many respondents belonged to Gen Z. If such data is unavailable, please show it as a study limitation. The article is interesting. But please extend the description of the conducted research by clearly indicating: the research gap, research goal, and research problems. The method of selecting the sample for research should be described in more detail. For example, were the respondents random?;  how were they selected for the research? How was the research literature selected? Finally, in "Conclusions", it is necessary to indicate what was confirmed by the research, how the research problems were solved?; what the research limitations are. (There is nothing wrong with the fact that researchers indicate study limitations. The research will never be able to analyze everything. However, it also creates an opportunity to indicate what, in the authors' opinion, the following areas of research should be pointed out and why).

Author Response

Dear reviewer,

I hope you are doing well. Please see attached as the response letter to your valuable comments.

Thank you for your effort and help.

Round 2

Reviewer 2 Report

No comments. Great job! Thank you 

Author Response

Dear reviewer,

Thank you and we appreciate your effort in helping us.